# How to implement a clinical ethics committee in an oncological research hospital: Qualitative results from a process evaluation study using normalization process theory (EVACEC)

Marta Perin[1], Morten Magelssen[2], Chiara Crico[1,3]*, Luca Ghirotto[4], Marco Annoni[5,6], Giorgio Gualandri[1], Ludovica De Panfilis[7,8]

**1** Legal Medicine and Bioethics, Azienda USL-IRCCS di Reggio Emilia, Reggio Emilia, Italy, **2** Centre for Medical Ethics, Institute of Health and Society, University of Oslo, Oslo, Norway, **3** Fondazione IRCCS Istituto Nazionale Tumori, Milan, Italy, **4** Qualitative Research Unit, Azienda USL-IRCCS di Reggio Emilia, Reggio Emilia, Italy, **5** National Research Council of Italy, Roma, Italy, **6** Fondazione Umberto Veronesi, Milano, Italy, **7** Dipartimento di Scienze Mediche e Chirurgiche, Alma Mater Studiorum, Università di Bologna, Bologna, Italy, **8** IRCCS Azienda Ospedaliero-Universitaria di Bologna, Bologna, Italy

\* chiara.crico@ausl.re.it

## Abstract

### Introduction

The Clinical Ethics Committee (CEC) of the Local Health Authority (LHA) of Reggio Emilia, Italy, is a multi-professional service established in 2020 to support healthcare professionals (HPs) in dealing with ethical issues in clinical practice. We evaluated the integration of the CEC into routine practice, 24 months after its implementation.

### Methods

We conducted semi-structured interviews with CEC members and LHA stakeholders involved in the service implementation. The interview scripts were outlined and transcript analysis was carried out following the four concepts of Normalization Process Theory (NPT): coherence, cognitive participation, collective action, reflexive monitoring.

### Results

Between June 2022 and January 2023, 15 participants were interviewed (12 CEC members and 3 LHA directors). All participants consider the service an important opportunity for HPs to be supported in complex situations (coherence). The CEC's President, a bioethicist working at the LHA, played a key role ensuring the CEC's participation and activation (cognitive participation). The main barriers to the CEC implementation were: financial sustainability, CEC members' lack of training, absence of in-person relationships (collective action). Overall, participants reported a positive

**Data availability statement:** The descriptions of the participants' experiences, vital for our study and object of our analysis, may contain references to the participants' personal identity. As such, these files cannot be shared publicly with article readers due to ethical restrictions. The point of contact for all data requests is our Research Ethics Committee. Here are their email addresses: CEReggioEmilia@ausl.re.it; SegreteriaLocaleCEAVEN@pec.ausl.re.it.

**Funding:** This study was partially supported by Italian Ministry of Health–Ricerca Corrente Annual Programme 2025. No additional external funding was received for this study.

**Competing interests:** The authors have declared that no competing interests exist.

experience with the CEC, however recommended several modifications (reflexive monitoring).

## Conclusions

We identified key components to support the normalization of CECs and enable their activation within a clinical setting. An active and sustainable CEC must be visible, accessible, understood and trusted, clear in purpose, sufficiently integrated into the life of the organisation, adequately resourced, appropriately constituted and competent, accountable and independent. These findings can inform the development of practical strategies for CECs implementation and of appropriate outcomes for further evaluation.

## Background

In response to the ethical issues and needs of patients, their families and healthcare professionals (HPs), clinical ethics support services (CESS) have increasingly been implemented over the past 30 years [1,2]. CESS are resources and frameworks generally established within healthcare institutions and designed to discuss ethical issues in clinical practice, support HPs in dealing with complex ethical decisions, and promote educational initiatives on ethical issues [3,4].

CESS are generally provided by either ethics consultants (in team or individually) or clinical ethics committees (CECs) [5]. Despite their growing presence in clinical settings, the usefulness of these services is still an object of debate [6], along with the appropriate ways of evaluating them [7].

One possible source of disagreement concerns the mechanisms through which CESS operate, as well as the metrics needed to measure their impact. According to Schildmann et al., little conceptual methodological work in CESS evaluation research has been conducted so far, leading to a current gap in normative and empirically informed evaluation of CESS outcomes [8]. Since CESS – and CECs – represent a complex intervention [8–10], a proper evaluation would require an understanding of how they operate, paying particular attention to their assumptions, the active components that influence the outcomes, and interactions between their elements [11].

Differently from previous studies on CESS implementation, which are mainly aimed at collecting potential users' needs [12–14], perspectives [15,16] and availability [17,18], perceived barriers and facilitators [16,19,20], or at comparing legislation [21], the methodological approaches and tools of health service research may provide new perspectives on this topic and increase knowledge about what it means to offer "good" CESS [8]. Health service research tools provide a comprehensive investigation on the processes and complexities of the service implemented, answering questions beyond efficacy and effectiveness [11].

In Italy, a national CESS regulation is still missing, despite the intense ongoing discussion about the opportunity to establish CESS to support HPs and to integrate them in the National Healthcare system [22–26]. Only a few regions (4 out of 20)

and a limited number of healthcare facilities have dedicated CESS [23,27,28]. Among these, the LHA of Reggio Emilia established two different services: (a) a Bioethics Unit; (b) a clinical ethics committee (CEC).

The LHA of Reggio Emilia is a local public health service providing health and social care, hospital services and primary care. It consists of six healthcare districts and six hospitals (one for each district). In addition, a Scientific Institute for Research, Hospitalization and Healthcare (IRCCS) in Advanced Technologies and Care Models in Oncology is incorporated into the Reggio Emilia hospital. A total of 1500 beds are provided by the LHA, with 180 beds dedicated to oncology patients.

The two LHA CESS are:

(a) The Bioethics Unit, which is a research unit, established in 2016 with the aim of promoting evidence-based ethics through research activities, ethics consultation and clinical ethics, and training programs. The Bioethics Unit operated as the only ethics support service until the CEC was established. An ethicist was available to support HPs through facilitating ethics reflection. A deeper description of the Bioethics Unit activities and their impact on clinical practice has been provided elsewhere [29]. The experience gained by the Bioethics Unit led to the hypothesis that HPs could also benefit from the support of a multidisciplinary institutional body (a CEC) in managing complex moral situations.

(b) The clinical ethics committee (CEC), which was promoted by the Bioethics Unit and implemented in 2020, with the aim of supporting HPs and local directors in managing ethical issues in complex cases and situations [10]. The CEC was composed of 15 permanent members, representing different professionals involved in the clinical decision-making process. The CEC may then call-in external consultants whenever needed (e.g., specific specialist, religious representatives, etc). The CEC was expected to (i) evaluate and discuss clinical cases whose management raises ethical concerns in clinical practice (ethics consultation); (ii) propose and conduct ethics training programs for healthcare professionals and the community (ethics education); (iii) analyse general moral problems related to clinical practice and developing policies on moral issues (policy development). During the first years of activities (July 2020–2022), the CEC held 22 meetings; 11 ethics consultations; 5 webinars (10 hours) on ethics consultation for HPs; 1 online seminar on clinical ethics committee and ethics consultation; and published 3 institutional policies on ethical issues [30,31].

Both services have been implemented as research projects within the IRCCS, which is dedicated to oncology. However, since ethical issues can arise in all clinical settings, they are in fact aimed at all professionals working within the LHA.

Connected to the implementation of the CEC, the Bioethics Unit has developed a research project called EvaCEC (its research protocol has been published elsewhere) [10]. EvaCEC is a mixed-method research project aiming at examining how to successfully implement a CEC in a hospital setting. The project builds on employing Normalization Process Theory (NPT) [32]. NPT is an action theory developed to identify and explain mechanisms of processes and factors that might promote or inhibit the routine incorporation of complex interventions into clinical practice. [32]. The project integrates quantitative and qualitative methods to identify crucial variables of the implementation process, in order to get an accurate appraisal of the factors required for a CEC to become a well-established part of the hospital system [33].

This article will be the first in a series. It will introduce the qualitative results and discuss them, focusing on the experiences of actual CEC members and managers who contributed to the CEC's set-up. The experience of CEC users and local HPs will be addressed in separate publications.

## Methods

### Design

This is a qualitative study with semi-structured interviews, which were drafted and analyzed based on the NPT framework. NPT emphasizes four core constructs (see Table 1):

**Table 1. Core NPT concepts and related sub-concepts, downloaded from the NPT online toolkit [35].**

| CONCEPT | KEY ATTRIBUTE | WORKING DEFINITION |
|---|---|---|
| **Coherence** *Sense-making* The extent to which individuals understand all the elements of the intervention and the reasons for adopting a new intervention | Differentiation | Whether the intervention is easy to describe to participants and whether they can approach how it differs or is clearly distinct from current ways of working |
| | Communal specification | Whether participants have or are able to build a shared understanding of the aims, objectives, and expected outcomes of the proposed intervention |
| | Individual specification | Whether individual participants have or are able to make sense of the work-specific tasks and responsibilities – the proposed intervention would create for |
| | Internalization | Whether participants have or are able to easily grasp the potential value, benefits and importance of the intervention |
| **Cognitive participation** *Engagement* The extent to which individuals believe in the innovation provided by the intervention and start to prepare for it | Initiation | Whether or not key individuals are able and willing to get others involved in the new practice |
| | Enrolment | Whether or not participants believe it is right for them to be involved, and that they can contribute to the implementation work |
| | Legitimation | The capacity and willingness of participants to organize themselves in order to collectively contribute to the work involved in the new practice |
| | Active action | The capacity and willingness of participants to collectively define the actions and procedures needed to keep new practice going |
| **Collective action** *Enacting* What happens when the intervention is operationalized | Interactional workability | Whether people are able to enact the intervention and operationalize its components in practice |
| | Relational Integration | Whether people maintain trust in the intervention and in each other |
| | Skill Set workability | Whether the work required by the intervention is seen to be parceled out to participants with the right mix of skills and training to do it |
| | Contextual integration | Whether the intervention is supported by management and other stakeholders, policy, money and material resources. |
| **Reflexive monitoring** *Appraisal* The act of keeping an innovation under review and of adapting it intelligently to changing circumstances | Systematization | Whether participants can determine how effective and useful the intervention is from the use of formal and/or informal evaluation methods |
| | Communal appraisal | Whether, as a result of formal monitoring, participants collectively agree about the worth of the effects of the intervention |
| | Individual appraisal | Whether individuals involved with, or affected by the intervention, think it is worthwhile |
| | Reconfiguration | Whether individuals or groups using the intervention can make changes as a result of individual and communal appraisal |

1. Coherence: how participants understand the purpose and value of the intervention.

2. Cognitive Participation: the extent to which individuals are engaged and committed to the intervention.

3. Collective Action: the practical work required to implement and sustain the intervention.

4. Reflexive Monitoring: how participants evaluate the intervention's effects over time

We reported the methodology for this study according to the COREQ checklist [34], and a detailed description of the NPT core construct attached with the manuscript *Supporting information* (S1 NPT concepts in S1 File, S2 COREQ checklist in S2 File).

## Sampling and recruitment for interviews

The eligible population comprised professionals contributing to the CEC's development and implementation. We proposed the study to all the members of the CEC at the time and purposely to local health directors at the LHA of Reggio Emilia who contributed to the CEC's set-up. Only permanent CEC members were eligible.

### Ethics approval and consent to participate

The Ethics Committee of the LHA of Reggio Emilia (CE Aven) evaluated and approved the study (AUSLRE Protocol n° 2022/0026554, 24/02/2022). The protocol has been registered on ClinicalTrail.gov (registration n. NCT05466292). Written informed consent was obtained from all participants after a clear explanation of the study objectives and to ensure confidentiality. All methods were carried out in accordance with relevant guidelines and regulations.

### Data collection

A series of one-to-one interviews was conducted to understand stakeholders' perspectives on the CEC implementation process. Separate topic guides were developed for each stakeholder group (CEC members and LHA directors); both the interview topic guides have been informed by the four NPT concepts [32] (Table 1). The interviews guides were developed by the PI (MP), a PhD candidate who also acted as the CEC Administrative Secretary, and revised by two team researchers (LG and LDP). MP performed a pilot interview with two participants; small adjustments were made to the interview guide, but no major changes were necessary. The interviews were conducted by the PI and a team researcher (MP and CC). MP, as CEC secretary, had a strictly professional acquaintance with all participants. CC, being external to the service, had no prior engagement with most of the participants. CC and LDP are female healthcare researchers in bioethics with a PhD and a background in qualitative research. LDP was also the CEC president at the time of the study. LG is a male PhD and the head of the qualitative research unit.

Key areas covered in the interviews were

- perception of the role, utilization and impact of the CEC in the local context and reasons for implementing the CEC;

- participants' understanding of own activities in support of the CEC;

- evaluation of participants' own experience in terms of perceived support, impact on the clinical case under study and daily work;

- suggestions and overall opinion of the CEC.

Both the interview topic guides are reported in *Supporting information* (S3 Interview topic guides in S3 File). Participants were invited by an e-mail sent by the PI. The interviews were conducted online, audio-recorded with the participant's written consent and transcribed verbatim. All interviews were conducted in Italian, participants' mother tongue; the quotes used in this manuscript were translated into English by the authors. No repeat interviews were conducted, and the transcripts were not returned to the participants for comments or corrections. Data collection began on June 24th, 2022, and was completed by January 26th, 2023.

### Data analysis

Qualitative data were analysed using the NPT toolkit [32] and framework thematic analysis [36]. Qualitative analysis was performed by research team members (MP and CC) with a deductive/inductive approach. In the deductive stage, the texts were categorised according to the NPT constructs on Excel files (version 240). After this initial categorisation, an inductive analysis was conducted within each NPT construct. This involved coding individual text segments with descriptive labels. Subsequently, these codes were used to identify and define overarching themes and subthemes (S4 NPT codes in S4 File). The identified themes and subthemes were further organised into categories of barriers, facilitators, and suggested improvements. Data were finally reported according to the NPT core constructs and identified by a specific code, composed as follows: numeric reference of the participants (01, 02…); reference to the stakeholder category - "CM(s)" for CEC Member(s), "LD(s)" for Local healthcare authority Director(s). Any disagreement between the two researchers who performed the analysis was resolved by discussions, until consensus was reached. Results were discussed among all the researchers and confirmed.

## Results

We invited 16 prospective participants. One of the CEC members did not respond. The final sample consisted of 15 participants: 12 CMs (3 bioethicists; 1 palliative physician; 1 anaesthesiologist; 1 nurse; 1 legal physician; 1 pharmacist; 1 paediatrician; 2 legal expert; 1 citizen's representative); 3 LDs (research institution operating Director, the health Authority General Director; the Director of research infrastructure). Further characteristics are listed in Table 2. Notably, while LDs had no experience in bioethics, CMs had different, sometimes combined, experiences in bioethics training. 8 CMs had received specific training, such as master's degree or PhD; 5 reported to had gained bioethics competencies through clinical experiences; the majority of CMs [9] reported to have had other professional experience, such as being a member of an Institutional Review Board or having performed voluntary work.

Interviews took place between June 2022 and January 2023 and lasted an average of 41'00" (min 27'00"- max 52'35"). Results are presented structured by the four core NPT concepts. Final results are also described in Fig 1.

### COHERENCE – "*An opportunity to deal with ethics in clinical practice*"

All participants identified providing ethical support as the CEC's primary goal. The CEC can provide support mainly by "*giving personalised, multiprofessional and competent answers*" to HPs' concerns whenever they experience ethical conflicts, and by being "*at the HPs' side*" when they must make difficult decisions (01-02CM; 04CM; 07CM; 09-10CM; 14CM; 02LD). To some participants, the CEC represents a crucial opportunity towards what they called the "humanization of medicine", that is, contributing to a cultural change in the health care approach. This would happen by considering the ethical aspects of care and patient values as part of the decision-making process (05CM; 14CM) by: (a) making relevant doubts and uncertainties explicit (04-05CM; 08CM, 14CM); (b) providing a space to reflect on the value of medicine itself (02LD); (c) educating HPs and citizens on healthcare ethics (04-05CM; 10CM; 14CM).

> *I think the CEC's goal it's partly (…) humanizing a little bit… (…) but also precisely increasing the quality of care perceived by people. I feel the one who brings the point of view of those who might be affected by these decisions. [14CM]*

All CEC members identified HPs as the main intended users of CEC services, though only a few identified patients and the general public as possible target audiences (13-14CM). The CEC was envisioned as a space for debate, dialogue, and in-depth analysis of moral concerns. One of the CEC objectives is introducing an ethical approach to clinical decision-making by stimulating questions that HPs seldom ask themselves (08-10CM; 3LD) or addressing problems that HPs have not been prepared nor trained to manage (02-03CM, 03LD).

**Table 2. Demographics characteristics of participants and CEC members' experiences with bioethics.**

| | *CEC Members (CMs)* | *Local healthcare authority Directors (LDs)* |
|---|---|---|
| *n. of participants* | 12 | 3 |
| *female* | 5 | 3 |
| *male* | 7 | 0 |
| *30 < age < 40* | 2 | 0 |
| *41 < age < 50* | 3 | 3 |
| *51 < age < 60* | 3 | 2 |
| *61 < age < 70* | 3 | 0 |
| *70 <* | 1 | 0 |
| *Specific training in bioethics* | 8 | / |
| *Fieldwork experience* | 5 | / |
| *Other professional experiences in bioethics* | 9 | / |

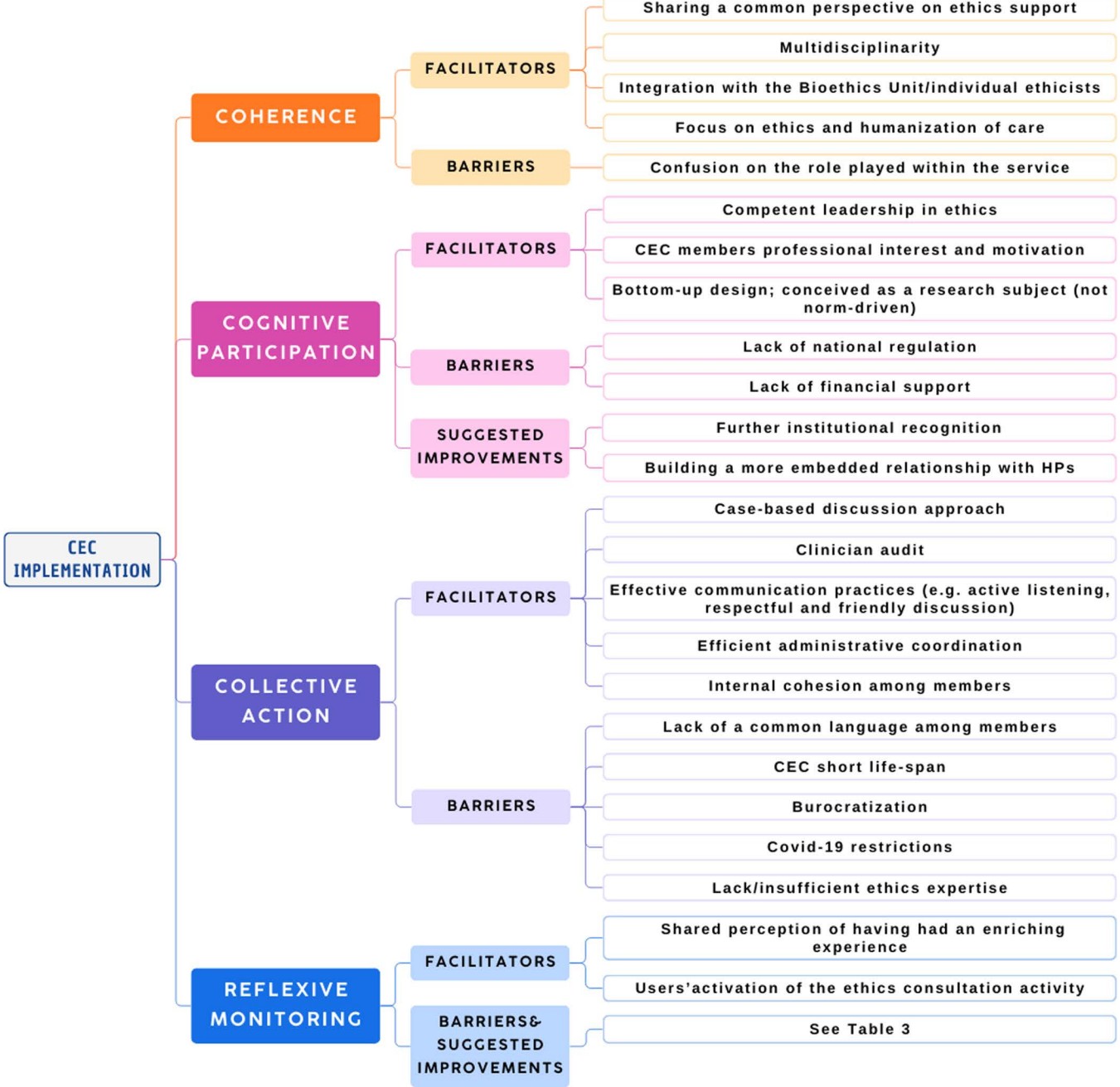

**Fig 1. Results summary according to NPT core concepts: this figure summarizes the results of the NPT core concepts analysis, in terms of emerging facilitators, barriers and suggested improvements.**

*Our task is not to explain to the physician their professional responsibilities, (but) to help them to reason about what is good and what is bad for the patient at that moment, in that given context. [10CM]*

*In the moments of greatest tension, I think (the CEC) could help us refocus a little on the value of medicine, life, and the end of life. [02LD]*

As for their role within the CEC, CMs described it in different ways: some perceive themselves as "guarantors" of the interests of those who will be affected by the decisions (HPs, patients, relatives, caregivers) (02CM; 09-10CM; 14CM):

> At least, I would like to help them (HPs) in reasoning and making the situation they're facing a little clearer, thus helping them serving/acting in the patient's interest. [02CM]

Some emphasised their contribution to the CEC by providing their academical expertise in bioethics (08CM; 10CM) or ensuring, as "interested stakeholders", that the CEC has the necessary budget to carry out its activities (01LD; 03LD).

Participants perceived the CEC as novel and innovative compared to local and national practice (07CM; 09CM; 14CM; 02LD), mainly due to:

i) *Multi-disciplinarity*, which "brings people with completely different background together" (13CM). The CEC would facilitate the collection of important information and allow for different competent considerations, thus supporting physicians in providing more appropriate recommendations to patients (01-02CM; 04-05CM; 01LD; 03LD).

ii) *The focus on holistic care* implies a broadening of the horizon beyond clinical and legal assessment, including social, relational, and spiritual issues (02CM; 05CM; 07CM; 09CM; 14CM; 03LD).

> It's something very different, that can be extended beyond the hospital or health care (…) I think it's a view: looking beyond the health aspects, and this is perhaps absolutely the biggest specificity. The main institutional body of a hospital usually limits itself to the medical-health aspects, and with the CEC there are opportunities to stretch towards the spiritual, religious, social, relational, cultural aspects of care. [09CM]

iii) *Integration between the CEC and the Bioethics Unit*: according to some CMs, the differences between the two services and their possible integration would provide a "*comprehensive ethical service*" to make healthcare professionals more aware of the ethical aspects of their clinical practice (01CM; 08-09CM). For other participants, while the Bioethics Unit provides a more personal relationship, the CEC is an "institutionalized" support, leading to a "*different way to deal and solve ethical problems*" (04CM; 08CM), as it deals with more complex, conflictual, ethical cases which can be properly addressed by a multidisciplinary expert team (01CM; 05CM; 08CM; 10CM; 03LD;).

**COGNITIVE PARTICIPATION – "*Institutional and individual resources to bring the CEC into practice*"**

Reflecting on the establishment and functioning of the service, participants unanimously acknowledged how crucial the Head of the Bioethics Unit – the CEC president – was in all the steps of the process: from selecting and involving CMs in the committee activities (07-10CM; 02LD) to managing all CEC operations. CMs report that a competent president should: "*be experienced in clinical bioethics*", "*hold an institutional role but also an operative role*" (01CM) and provide CEC members with a method to deal with ethics consultation and ethical analysis (01CM; 08CM; 10CM; 14CM). Competent leadership emerged as a key element for the CEC sustainability over time (04CM):

> We really need someone who guides us and gives us the tools to discuss and deal ethically. This is crucial. [04CM]

According to the LDs and the CEC President, who were directly involved in establishing the CEC, the CEC was not implemented in order to fulfil external demands or criteria set by overarching institution - at national or European level - (01LD) but as a research project aimed at studying the approach to ethics consultation in a direct relationship with HPs (01CM; 01LD). According to two Directors, the experimental character led to an implementation process in which the CEC emerged as "*something built by HPs, for HPs, and with HPs*" – which they called bottom-up approach (01–02LD).

 

Consequently, the CEC's experimental character, from both a bioethical and organizational research perspective, emerged as a distinctive feature (01LD; 07 CM), prompted by the CEC being embedded in a Research Oncological Hospital (01–03LD) and facilitated by the scientific experience and expertise of the Bioethics Unit (01–02LD):

> The thing that impressed me the most in this process is that the whole thing was born from the bottom—bottom/up, not top/down. So, it was born from the professionals. By "the professionals," we mean the professionals working in palliative care within the oncological research hospital. [02L]

From the CMs' point of view, curiosity (02CM; 07CM; 13CM), personal and professional interest towards medical ethics and innovative services (04-05CM; 07-11CM; 14CM), the prospect of personal and professional growth (02-03CM; 08CM; 14CM) were driving motivations to joining the CEC. Additionally, participating in the CEC was seen as an opportunity to gain training and to self-educate in bioethics (03CM). Two CEC members added as further motivation their desire to be "useful" and help other colleagues (02CM; 07CM). Moreover, CMs described themselves as volunteers since they do not receive attendance fees and describe their participation as highly time-consuming (01CM; 04CM; 07-09CM).

> I do a job where the bioethics component in my opinion is quite heavy and I haven't any specific training, except for on-the-job training as HPs, which is derived from the fact that you're in it. So, in my opinion, it's also a way to broaden my horizon of views. [03CM]

> The desire to make myself useful was the levers that prompted me to accept this invitation. [07CM]

Considering its potential, some participants underlined the importance of maintaining and promoting a clinical ethics support service within healthcare organisations (01CM; 04CM; 08CM; 13CM; 03LD). Participants emphasised the need to find strategies to guarantee further institutionalisation, greater consistency and continuity, and recognition of this type of service with a national law (01CM; 07-08CM; 10CM; 14CM; 03LD).

Among these, economic support would '*'allow the service to make a qualitative leap*" (13CM), whereas scarce or absent economic support to CEC components and activities may jeopardize the professionalism and timeliness of its work (01CM; 04CM; 08-10CM).

> But the CEC cannot be an object of volunteer work. It is a job. First, because of the management of the data we come into contact with: you have access to patients' data, inevitably. So already, this cannot prefigure it as a volunteer activity. So, it is a real work activity, and as such, it needs contribution for the commitment, which is the meetings and back/office activities. [01CM]

According to our participants, selected strategies can improve the CEC activation: implementing education and training activities for HPs to provide them with answers and tools usable in their daily activities (04-05CM; 08CM; 03LD); building "*an almost daily relationship*" (11CM) with HPs at a higher risk of encountering ethically complex cases daily (e.g. those working in ICU, Neonatology, etc.) (09CM; 03LD); promoting the CEC and its visibility within the institution through organised communication initiatives (01-05CM; 10CM; 14CM), e.g. participating in interdepartmental (03LD) or multidisciplinary groups of pathology meetings (07CM). Two participants recognised including the CEC in the Legal Medicine Department of the LHA of Reggio Emilia, under its HealthCare Directorate, as a step forward in institutionalising the CEC (01CM; 02LD).

**COLLECTIVE ACTION (CA) – "*The experience of building ethical discussion*"**

Participants described their experience in dealing with clinical ethics – the working mode of the CEC – as "*the interweaving between 'principles theory' and concreteness of the case*" (10CM), emphasising the value of actual case-based discussion (01CM; 03CM; 09CM-11CM; 14CM).

As part of the CEC's working methods, healthcare professionals who seek its support are invited to participate in the case discussions. Their active involvement in the hearings, including sharing their perspectives and concerns, helps enhance the consultation process by fostering clearer communication and collaboration (01CM). This way, the CEC would represent a "*space that allows freer interactions between HPs and experts, who are not necessarily employed by the same institution, avoiding possible conflicts of interest and hierarchy which are inherent of services that are structured within the same facility*" (08CM).

Ethics discussions in the CEC were perceived as an "*enriching experience*", characterised by "*listening and mutual understanding*" (13CM), and an "*open, respectful and friendly discussion*" (08CM), which helps CEC members to grasp "*a non-partisan understanding of the problem*" (04CM). Some underlined certain features as key to their experience, such as "*reasoning*" and "*clarifying*" the difficult situations healthcare professionals face (02CM); listening, understanding, and sharing their own opinion and perspective, even when in contrast with other members' (08CM; 11CM). CEC members can indeed be "*on different wavelengths*" (03CM), but working together requires them to continuously balance their expertise, competencies and perspectives with the other members' (02CM).

Some CEC members recognized personal and professional limits, due, mainly, to their inexperience with ethics, ethics consultation, and more generally to their lack of ethics or bioethics training. In particular, participants felt poorly equipped to manage ethical aspects of care in individual complex cases, and to discuss such cases with their colleagues; this was a common perception especially across clinicians and citizens' representative (02-04CM; 07CM; 09CM; 14CM). As reported by one CM, a lack of a '*common language*' can impair mutual understanding and the implementation of ethical deliberation (02CM).

Despite their doubts and perplexities, CMs perceived that they could contribute not primarily by applying their individual professional knowledge and competences but rather by applying a method of discussion (03-04CM; 07CM; 09CM):

> *The fact that the first case presented was from pediatrics also helped me, in that I managed to deploy my personal expertise. Then I realized that it was about a method, and not about a specific expertise. [09CM]*

The skills of the individual and the overall work of the multidisciplinary team are reported as pivotal elements to build the accountability of the service (01LD).

Considering the contextual integration, the internal cohesion among CMs, the organization provided by the administration office and the coordination supplied by the President were identified as crucial aspects for the CEC's successful implementation (08CM; 14CM).

However, many CMs perceived the CEC has an undeveloped potential for HPs and that several barriers may still negatively affect a wider use of the CEC (02CM; 8CM; 11CM; 14CM). Among these: the"*bureaucratic aspects*" involved in making a formal request to the CEC (01-02CM; 14CM), the lack of personal interaction with HPs and among CMs due to Covid-19 restrictions (04-05CM; 10CM; 13CM); the lack of knowledge and/or understanding of the service itself and its scope (09CM); poor awareness of and cultural attitudes towards ethical issues among healthcare professionals (02CM; 04CM; 09-10CM; 03LD). Particularly, participants noted that a comprehensive embedding is thought to require a longer time than 2 years (02-03CM; 09-10CM; 13CM).

> *Generally, in my opinion, there is hardly any awareness of ethical issues. Professionals don't have these matters clear in their minds. It's not that they have no idea about it, they don't know how to bring up these issues…[04CM]*

Education targeting employed HPs was identified as important to raise awareness about the CEC and strengthen the relationship between the CEC and the institution (04CM; 10CM) and build a common level of ethical awareness (02LD). Communication efforts have to be improved, as the initial disseminating events were perceived as insufficient to efficiently raise awareness about the CEC (01CM; 08CM). Despite several efforts, the message still needs to improve (01CM).

*It is the CEC that has to adapt to professionals, and not vice versa. (…). If it is perceived as a formal institution that is far from everyday reality, you have to ask yourself why. [01CM]*

**REFLEXIVE MONITORING – "A promising work in progress"**

Generally, participants felt satisfied with their experience and considered the CEC a successful project, with promising results (01-02CM; 07-09CM; 11CM; 13-14CM; 02-03LD). They believe they have learnt a new "*way of relating*" to ethical issues (13CM) and gained an enriching experience for their profession (01CM; 04CM; 07CM; 14CM). CEC components described the CEC as an "*added value*" (05CM; 07CM) for the hospital:

*It's an enlightened choice, for an organization which organizes healthcare services at a territorial level, that is not only concerned with bare organizational and clinical aspects, but also precisely with everything that can help to keep the system running at its best. So obviously a CEC can be of great help and support, as I said before, both to individual professionals and citizens. [07CM]*

*Beyond that I see that, meeting after meeting, I personally learn a lot of things, not just at the level of content or issues that in many cases I had never dealt with, I think I'm learning this way of relating. [13CM]*

Ethics consultation – and the related discussion on individual clinical cases – is for many CMs "*the most valuable*" (09CM) and high-impact activity provided by the service (01CM; 03-04CM; 08-09CM): this work is what makes the CEC useful (08CM).

Few participants evaluated – or even talked about – the remaining tasks of the CEC: the educational activities and the development of ethics policies. The first one was only mentioned by one CM, who gave a positive evaluation (10CM). About policy development, two CMs reported that, while useful in specific circumstances (e.g., during the pandemic), this activity was time-consuming and possibly had little impact on everyday clinical practice (01CM, 08CM). According to others, it can be valuable to feed the public debate and foster a cultural awareness on sensitive ethical issues among HPs and citizens (01LD, 09CM) but is not perceived as useful by healthcare professionals nor health management (01CM).

Timing emerged as a crucial factor in providing ethics consultations: for some, the time needed for CMs to meet, discuss the case and produce an opinion does not fit the time required by cases in clinical practice, thus jeopardising the service's usefulness (03CM). On the other hand, a longer time to provide an opinion was sometimes perceived as a strength, as it allows a better evaluation of the clinical case since the multidisciplinary of the CEC itself hindered the possibility of an immediate reply to ethical concerns (03CM).

*However, I don't think this is a critical issue, it is a fact; I also believe that a committee, which is composed of fifteen people cannot give an answer in a day or even in two or three days afterwards; it is in the very being of a committee struggling to provide such an answer. [05CM]*

Participants provided several suggestions for modification and, occasionally, reorganization of the CEC, in order to improve its integration, as reported in Table 3.

According to the participants, the CEC of the LHA of Reggio Emilia might represent a benchmark for the implementation of similar support services (01CM; 05CM; 01LD; 03LD). Yet, transferability depends to a large extent on any similarities across local contexts, which may instead have very different social, political and economic characteristics (3LD). According to a participant, these inherent differences would make any strict any strict CEC standardisation unfeasible; on the other hand, the chance of very different, even incompatible, interpretations of the same ethical question should be carefully monitored (07CM).

**Table 3. Suggested modification/improvement to current CEC operation, as identified by participants (Reflexive Monitoring).**

| SUGGESTED TIPS | DESCRIPTION |
|---|---|
| *Planning in-person meetings*<br>*Equipping the CEC with an office* | Both are strategies to ensure a *physical presence* of the CEC and personal relationships within the institution (02CM; 04CM; 13CM).<br>Online meetings are easily accessible and useful (04CM; 10CM; 13CM), but sometime perceived as a limitation for discussion, confrontation and CEC integration within the healthcare facility (04CM; 13CM). |
| *Improving reciprocity by in-person communication and feedback assessment* | Suggested actions to improve feedback and reciprocity with CEC potential users:<br>•rethink the moment of providing HPs with a written opinion as to improve relationship with the requestor (e.g., the opinion may be delivered in person) (02CM).<br>•the opinion may be shared with the treatment team, as a moment of further education on case-specific ethical aspects (instead of a 'one-to-one dialogue') (02CM; 04CM).<br>•ask for HPs' feedback on the activities promoted by the CEC (ethics consultation and ethical policy development particularly) (04CM). |
| *Simplifying and expanding the EC request procedure* | The written, online submission of the ethics support requests has been perceived by CEC members as restrictive of the complexities involved (13CM).<br>Suggested strategies include expanding to all CMs the ability to refer requests for EC to the CEC (01CM) and implementing a structured format clarifying the ethical concerns to simplify the EC request submission (13CM). |
| *Featuring CMs' in-house training* | CMs should be provided with:<br>•self-training on bioethics and clinical ethics, both at the start of the term and during;<br>•a rigorous methodology for ethical analysis (01-02CM; 04CM; 8CM; 14CM);<br>•facilitation in understanding the issues discussed during CEC meetings. |
| *Creating a CESS/CECs network* | The creation of a national network of CECs was identified as a promising action to exchange opinions, good practices and build strategical relationships among professionals involved in CESS (01CM; 04CM; 07-10CM; 03LD). |
| *Adding extra specialized components to the CEC* | It has been suggested to add new specialized members/stakeholders to the CEC (03-04CM; 14CM) in order to better fulfil any new tasks of the CEC, i.e., the possibility of the CEC being responsible for evaluating requests for assisted suicide (13CM). |
| *Ongoing research on clinical ethics* | Research has been identified as an added value to collect evidence on the service (01CM; 08CM; 14CM; 01LD; 03LD;) |

## Discussion

To our knowledge, this is the first attempt to provide a process evaluation study of a CESS implementation within the specific NPT framework [32]: our results place the practical experience of CMs and relevant stakeholders within a robust theoretical framework for implementation, which can enable normative conclusions on the practical work and resources needed to implement a CEC - the specific form of CESS we studied - successfully.

In line with expectations for this service [3,10] as for the'Coherence' concept, the CEC emerged as an opportunity to support HPs, providing them with a space of debate, confrontation and support in difficult ethical situations.

To describe the aim of the CEC, some participants used the terms "*humanisation of medicine*". This concept generally refers to a broader, holistic perspective of healthcare, for which therapeutic efficacy cannot be determined solely on the basis of clinical outcomes but also depends on the consideration of the purely ethical aspects of care: e.g., balancing ethical principles, respecting patient values and dignity, and promoting humanistic values such as listening and empathy in the therapeutic relationship [37]. In this respect, the consideration of ethical aspects of care emerged as part and parcel of HPs' work and a vehicle for improving the quality of care.

Interestingly, despite the inherently ethical nature of the "*humanisation of care*" concept itself [38], its connection to clinical ethics is not widely found in the literature; nor is the assumption that the humanisation of care is even a long-term goal or possible outcome of ethical support services. This interpretation represents a unique perspective and an original result of the present study, to the extent that we assume that this association depends on the socio-cultural context of our participants.

As for the resources and work needed to practically organize the CEC, its members, and its activities within the healthcare institution ('Cognitive Participation'), the presence of an operating Bioethics Unit played a central role in promoting the intervention. The Bioethics Unit, whose members are competent in clinical ethics and ethics consultation, enabled and actively supported the CEC implementation, also through several research activities concerning the CEC [10,29,31].

Despite several calls for action [22,39–41] Italy still lacks a national regulation on CESS [42], causing several discrepancies in CESS implementation throughout the country [43]. Regardless, a research approach can represent an opportunity to assess context-specific needs and adapt the intervention to local requirements and existing resources, both for countries without a regulation - like Italy [22] – and for those that already have longer-standing CESS by law – like Norway [9].

Institutional support emerged as another key element of 'Cognitive Participation', encompassing the broadening of ethics consultation standardisation, the deeper integration of the CEC into healthcare systems and, above all, the financial remuneration of CMs, as already suggested in literature [1,23,44]. Other studies described institutional support as providing CMs with necessary working conditions to fulfil the CEC mandate, including reasonable access to professional literature and education — which would enhance CMs' professionalisation —, adequate time to engage in ethics consultation, and freedom from institutional pressure [23,44,45]. Additionally, employing dedicated professionals with a relevant background (health, ethics or law) to assist the chairperson with administrative and support work should be considered [45].

The approach of ethical deliberation applied to case discussion and the integration of the CEC within the LHA were found to be key factors in the NPT domain of 'Collective Action'. Despite a generally positive perception of how the CEC performed its activities, the low level of both individual and shared ethical competencies among CEC members may have challenged how internal discussion and ethical deliberations have been conducted. Little ethics competence and expertise with ethics consultation are issues that have been pointed out before and may have a negative impact of both the outcomes of the deliberations and CEC users' satisfaction [4,46,47]. To overcome these issues, a specific training for all CMs should be a primary goal of a new CEC [48]. Ideally all CMs should hold general skills – such as personal aptitude, the ability to work in a group, being capable of compromise, ability to listen and tolerate uncertainty and disagreement, being religiously neutral, and the ability to form critical thinking [1,45,48] – but only a few members need to hold advanced knowledge and specific skills, such as identifying and analysing an ethical conflict, referring to relevant ethics documents/theories, guiding ethics facilitation, reporting the ethics deliberation, and coordinating the service [49]. Being aware of this distinction and focusing on the specific contributions every CM can give to the deliberation may prevent unpleasant feelings of inadequacy across CMs, reinforce their commitment to deliberation and to other activities of the CEC, such as educational activities and policy development [49].

Similarly to other Italian experiences [23], while CMs strongly believe in the value of the CEC for the whole institution, they struggle with the discouraging fact that the CEC is still poorly integrated and largely unknown among healthcare professionals. The CEC lacking a physical and live presence within the LHA, along with insufficient and/or ineffective dissemination activities, may have affected its integration within the institution, as previously suggested [1,50]. To improve integration, participants call for increased mutual, daily, relationship between the CMs and HPs. It must be noted that the number of requests to the CEC may also be influenced by the ethical literacy of HPs, as clinicians with more advanced ethical education appear to be more likely to request ethics support [1,51]. Education in clinical ethics, with practical case-based discussion and experience of ethical analysis, might help HPs develop basic ethical competencies and the ability to identify ethical problems [23,51,52]. Educational activities should be a continuous task of the CEC.

Concerning the observations and remarks on the overall experience ('Reflexive Monitoring'), our findings are consistent with existing literature on CECs. Despite reporting areas for improvement, CMs feel satisfied with the service [4] and consider ethics consultation the most relevant activity of the CEC [4]. With respect to the latter, we observed a diverging

understanding of the time needed for the CEC to respond to consultation requests. For some, it is a barrier to the full integration of the CEC; for others, it is a prerequisite for an adequate and multidisciplinary response to the complex issues addressed in the consultation request. While it is certainly true that the demand for a fast response is often dictated by the requirements of clinical practice, these reflections need time to mature. Hence, the difficulty of responding at short notice is not - or not only - a purely organizational problem, i.e., the time required for a group of professionals to get together to discuss; it is rather an intrinsic problem, stemming from the need to deliver a well-reasoned and substantiated opinion. In an institution with diversified ethics support services, the CEC and the Bioethics Unit, timing might be considered as an additional - though not conclusive - factor in the assessment of the type of need (urgent/non-urgent), to match the request to the appropriate service.

## Practical implications

Our analysis enabled us to identify the active components required to successfully implement the CEC. To better inform a normative and empirically sound process evaluation of CESS implementation, we provide a detailed description of each component along with its proposed outcomes and tools for their assessment in Table 4. Its application will require further investigation.

If some of our results are, as we hypothesized, likely due at least in part to the Italian sociocultural context, we believe that the active components for effective implementation are not particularly affected by cultural factors and can therefore be transferable to most contexts.

## Strengths and limitations

This is the first study which provides precise indications on how to implement CEC in the absence of national legislation within an Oncological research hospital. Consistent with the positive evaluation expressed by CEC Members and stakeholders, the Emilia Romagna Region has progressively shown a strong interest in the experience gained by this CEC. After our study, such interest led the Emilia Romagna Region to publish a resolution to turn the CEC into a service with regional responsibility, with the institution of a Regional Clinical Ethics Committee [53].

This study also has limitations. To grasp the 'normalisation' of the CEC and to understand how it might provide a cultural change within the institution, more time and repeated evaluation should be conducted over a longer period.

As explained in the Methods section, the study has been conducted by a PhD candidate who was also responsible for the CEC Administrative Secretary (MP) and developed with the help of the CEC president (LDP): while this allowed a deeper understanding of the processes, such involvement could also lead to several biases. To minimize bias in data collection and qualitative analysis, LDP was not involved from the interview conduction phase onwards and data have been analyzed independently by two researchers, one of which (CC) is external to the LHA CEC and never participated in its activities.

Although the study was conducted on a CEC implemented within a cancer research institute, no requests were received from the oncology field, although ethics consultation has been proved useful in helping HPs managing the many ethical challenges that may arise in cancer patient care. We hypothesize that the absence of ethical requests from oncologists could be attributed to several factors, such as a lack of awareness of the service, cultural perspectives towards end-of-life care, and ethical issues in clinical practice. We regard this as a significant aspect that warrants further investigation.

Representatives of minorities or religious groups are not permanent members of the CEC but may be called as external experts if the case requires it; however, we acknowledge that this may not be sufficiently representative of other cultures and diverse communities.

Lastly, as findings arise from the specific context studied, some of the findings might not be generalizable to all hospitals or CECs.

**Table 4. Descriptions of the Active components involved in the successful implementation of a CEC. The list and their description (Rationale) represent a re-elaboration of the results; the Outcomes and Assessment tools represent a proposal of the research team based on personal experiences, scientific literature, and insights from the EvACEC implementation study.**

| Active component | Rationale | Outcomes | Assessment tools |
|---|---|---|---|
| Specific approach in ethics consultation | Adopting a case-based methodology of ethical analysis and involving the HPs who required the ethics consultation could improve ethics consultations. | N. of ethics consultation performed according to a specific method of ethical analysis/total of ethics consultation performed | -CEC meeting minutes |
| | | N. of hearings of professionals who requested CE/ total consultations provided | -CEC meeting minutes<br>-Secretariat database |
| Competent leadership | CECs need a leader who is able to ensure good coordination between HPs, CMs and LDs; guide and facilitate ethics deliberation; write well-argued consultation responses. | •% time devoted to the CEC/ total monthly time;<br>•Having a specific background in clinical ethics; | -President's monthly activities report;<br>-President's curriculum; |
| Local awareness of clinical ethics | Providing ethics programs to improve HPs ethical sensitivity and training them to identify ethical issues in their clinical practice. | •N. and type of participants in the initiatives;<br>•N. of staff dedicated to bioethics activities (research, training, ethics support);<br>•N. of bioethics-related initiatives promoted by the healthcare institution (trainings, conferences, seminars); | -Survey on HPs' ethical sensitivity with timely follow-up;<br>-Institutional reports on any ethics program/ activities; |
| | Improving knowledge of the CEC and its activities among Department Directors | •N. of meetings for presenting/ updating CEC activities across departmental meetings<br>•N. of activities/requests for support on internal protocol procedures/revision | -Annual activity report (n. of protocols/guidelines on which CEC oversight was requested);<br>-Survey on Department Director's ethical sensitivity with timely follow-up; |
| Systematic evaluation system | Promoting activities aimed at service evaluation and its penetration in clinical practice. | •HPs' adherence to CEC's policies/ethics consultations;<br>•HPS' participation in CEC training activities; | -follow-up on ethics consultation outcomes and quality (with ECQUat scale)<br>-internal surveys;<br>-annual report of activities; |
| Institutional support | Support from the local healthcare institute ensures smooth operation, maintenance and integration of the CEC within departments. | •Financial sustainability;<br>•Provision of facilities for the CEC; | -Payment of attendance fees;<br>-In-house staff specifically assigned to the CEC; |
| Trained CEC members | Selecting professionals with different levels of skills in clinical ethics as CEC members is an element of quality assurance of ethical deliberation. | •Having received at least a specific training in clinical ethics; | -CEC members' curriculum;<br>-ACES questionnaire; |
| | | •Delivery of an annual in-house course to structure and increase the ethics consultation skills of CMs | -Active participation in in-house training;<br>-Pre-post training questionnaires; |
| Physical presence | Providing opportunities and/or spaces for the CEC to have a physical presence in the healthcare institution is expected to improve both CEC activation. | •N. of CEC activities delivered online/in-person;<br>•N. of CEC in-persons meetings; | -Annual CEC activities report; |

## Conclusions

Our work explored the implementation process of a multidisciplinary CEC within an oncological research hospital. We identified components and empirical mechanisms that contribute to its normalisation in clinical practice, along with the changes needed to improve the service and to support its activities within the healthcare facility. To be active and

sustainable, a CEC must be visible, accessible, understood and trusted, clear in purpose, sufficiently integrated into the life of the organisation, adequately resourced, appropriately constituted and competent, accountable and independent. After two years of operation the CEC at the LHA of Reggio Emilia partially fulfils these aspects, resulting in an overall positive evaluation by informants who report a positive experience, yet with wide room for improvement regarding its normalisation in the organization. CMs should be supported in the development of ethical skills and knowledge, while CEC projects and activities would benefit from LDs support in terms of financial sustainability, visibility and knowledge of the service and its potential for quality of care.

Further exploration and follow-up evaluation studies are needed to understand normalization in light of the emerging barriers and their resolutions and its supposed development over time. Further studies are also needed to explore implementation of CECs in other healthcare contexts.

## Supporting information

**S1 File.** NPT core concepts: this file contains the description of the concepts of the Normalisation Process Theory (NPT), the framework we used to inform both the interview guide and the analysis of our data.
(DOCX)

**S2 File.** COREQ checklist: it is the checklist used to report qualitative studies.
(PDF)

**S3 File.** interview topic guides.
(DOCX)

**S4 File.** NPT codes - English version: this is the file used for the qualitative analysis, which contains the text of the interviews (translated into English) labelled with codes, sub-themes and themes and the index of the analysis.
(XLSX)

**S5 File.** Interviews transcripts_raw data: this folder contains all the raw data used in this study, the interview transcripts (in original Italian).
(ZIP)

## Acknowledgments

The authors are grateful to all the CEC members who are dedicating their time to this service: Ludovica De Panfilis, Silvia Tanzi, Pierpaolo Salsi, Giancarlo Gargano, Monica Guberti, Giorgio Gualandri, Giulio Formoso, Sergio Amarri, Marco Annoni, Carlo Botrugno, Donata Lenzi, Roberto Satolli. Their collaboration has been invaluable, as is their commitment to bringing ethics to the patient's bedside.

## Author contributions

**Conceptualization:** Marta Perin, Morten Magelssen, Luca Ghirotto, Ludovica De Panfilis.

**Data curation:** Marta Perin, Chiara Crico.

**Formal analysis:** Marta Perin, Chiara Crico.

**Investigation:** Marta Perin, Chiara Crico.

**Methodology:** Marta Perin, Chiara Crico, Luca Ghirotto.

**Project administration:** Marta Perin, Chiara Crico.

**Supervision:** Morten Magelssen, Luca Ghirotto, Ludovica De Panfilis.

**Validation:** Chiara Crico.

**Visualization:** Marta Perin, Chiara Crico.

**Writing – original draft:** Marta Perin, Chiara Crico.

**Writing – review & editing:** Morten Magelssen, Luca Ghirotto, Marco Annoni, Giorgio Gualandri, Ludovica De Panfilis.

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
