## [Decision Letter · Decision Letter 0]

19 Nov 2024

PONE-D-24-40359How to implement a Clinical Ethics Committee in an Oncological Research Hospital? Qualitative results from a Process Evaluation Study using Normalization Process Theory (EVACEC).PLOS ONE

Dear Dr. Crico,

Thank you for submitting your manuscript to PLOS ONE. After careful consideration, we feel that it has merit but does not fully meet PLOS ONE’s publication criteria as it currently stands. Therefore, we invite you to submit a revised version of the manuscript that addresses the points raised during the review process.

We look forward to receiving your revised manuscript.

Kind regards,

Alejandro Botero Carvajal, MD

Academic Editor

PLOS ONE

Journal Requirements:

2. In the online submission form, you indicated that [Some data cannot be shared publicly due to confidentiality. The full text of the interviews would reveal our participants' identity. These data can however be shared with reviewers upon reasonable request].

Reviewers' comments:

Reviewer's Responses to Questions

**Comments to the Author**

1. Is the manuscript technically sound, and do the data support the conclusions?

Reviewer #1: Yes

Reviewer #2: Yes

Reviewer #3: Yes

Reviewer #4: Yes

Reviewer #5: Partly

2. Has the statistical analysis been performed appropriately and rigorously? 

Reviewer #1: Yes

Reviewer #2: N/A

Reviewer #3: Yes

Reviewer #4: N/A

Reviewer #5: N/A

3. Have the authors made all data underlying the findings in their manuscript fully available?

Reviewer #1: Yes

Reviewer #2: Yes

Reviewer #3: Yes

Reviewer #4: Yes

Reviewer #5: Yes

4. Is the manuscript presented in an intelligible fashion and written in standard English?

Reviewer #1: Yes

Reviewer #2: Yes

Reviewer #3: Yes

Reviewer #4: Yes

Reviewer #5: No

5. Review Comments to the Author

Reviewer #1: Thank you for allowing me to review this manuscript. In general, I really liked the topic and I believe in the importance of a CEC. I enjoyed how the authors followed the NPT framework and reflected on it to shape it into an adequate model to evaluate and normalize the CEC. It is a valuable, time-consuming and analytical work and I thank them for their effort to put together a nice work. However, I think the manuscript needs improvement in terms of English language, typing errors, some methodology and some structure failures.

Below are my comments:

Title: I suggest you remove the question mark as the structure of the title suggests an informative guide, so the question mark is unnecessary.

I was a bit confused whether the CEC is only for the Oncology consultations or not. You don’t introduce oncology until later in the practical implications but it is not clear if this is your aim. Line 69 and 70 you talk about oncology but then in the methods/results/discussion you don’t explain why and how oncology fits. I suggest you make a certain link clearly early in your manuscript.

Misspelling and punctuation:

Line 13 of an (the) Italian, line 27 relationship- missing an S, line 62 only few regions- missing an a in few: only a few, line 115 double point…

Please keep the same style consistently within your manuscript: the citations embedded within the text in the Results are sometimes italic sometimes not (i.e. line 162/163 and line 218/219)

When CM and LD is used in plural, the s is missing (i.e. line 205 seven CM)

Grammar:

Line 19 informed with, line 27 though- replace with but, however…, Line 40 a kind- replace with a type, a form, a part…, line 82 whose- replace with that, line 91 whose- replace with that, line 350 and 362there is on the one hand in the paragraph

Abstract

Line 23: maybe use another term than the general term stakeholder.

Background

Line 45/46, line 56/57, line 60/61: CESS are used CEC interchangeably. If CEC is part of the CESS why not focus first on the CESS and then move to focus on the CEC.

Line 70/71: it seems like a single statement. It should be better integrated in the paragraph. (might be a good link for the oncology part, see above my comment on oncology)

Methods

Missing: did you have a pilot interview after which you adapted the interview guide

Line 105: introduce a small definition of the steps of the framework NPT (what is coherence, cognitive participation…) in addition to the table.

Line 106: I suggest you include domain 1 and 2 of your COREQ in the methods rather than at the end of the manuscript, specially that you’re using a reflexive analysis, where the reflexivity of the authors and their influence should be clear and thought through from the beginning. I suggest you include a section on the reflexivity of the authors since they might have a relationship to the CEC, members of the CEC? (line 504: this is where you mention that MP works somehow for the CEC, this should be mentioned in the beginning and it should be reflected upon).

Line 114: you mean two separate one for the LDs and one for the CMs? clarify

Line 116/117: you mention PhD project and roles of the researchers later in the limitations section (line 503-403)

Line 119: some key areas are repetitive, can you combine some concepts together?

Line 130/131: clarify how did they get the names, is there a list?

Line 132: was the consent obtained verbally or written

Line 136: were the interviews done in English? Italian and then translated?

Line 142: supplemental material do you mean figure 1?

Line 147: restructure the sentence clarifying if there were conflicts of ideas between researchers and how they were resolved (after discussion, consensus was reached?)

Results

The results descriptive section and the figure 1 do not necessarily correlate. For example, in the coherence part, you enumerate i) multi-disciplinarity ii) focus on holistic care iii) different from the facilitators multidisciplinarity, integration with the BU and focus on ethics.

Line 224: you mention the CEC president, is this person not anonymous on purpose? I suggest you still name them like the other participants.

Line 274: I am confused which HPs and the case discussion? Is it a general statement or a part of your results?

Discussion

Humanisation of care/medicine is a very important finding indeed. You emphasized it beautifully.

I suggest you shorten the discussion by omitting a literature review on the topics that were not mentioned in your results section.

You mix here again the CESS and the CEC. Is it intentional? All your results are on the CEC if I understood correctly so why discuss the CESS?

Line 392/393: how did you come to this conclusion? Can you clarify? otherwise it is more of an assumption?

Line 421: Can you compare to your results? (similar to what you did one line 448 and 460)

Line 430: Interesting! Can you compare to your results? (similar to what you did one line 448 and 460)

Practical implications

Very important part of your research. Make sure you clarify if table 4 proposed outcomes stem all from your results or personal ideas and suggestions?

Reviewer #2: The evaluative work of any new establishment of new project is essential, hoever, some points requiered to be raised:

1- How the health care professional and the practitioner have managed to take care of ethical issues before the inception of this committee.

2- Who and how the need of such committee is decided and how it is regulated according to the Italian rules.

3- What about the other religions and immigrants, does this committee represent all the diverse community who uses health systems in Italy.

4- Do you think this short time of committee assessment is not necessarily a representative of this committee, may be this data used to improve the drawbacks and give the publication of the results more time, normally to change of the people could affect the action of it in the future. So, we can avoid the personal effect and detect more clearly the professional effect of such committee.

5- Does the speciality of oncological hospital require distinct ethics committee, may one regional committee could be enough and for all specialities.

6- There should be part of this work to the patients’ opinion about any improvements has occurred in the services offered.

7- Can the authors elaborate more about the competences of the committee members.

Reviewer #3: Review of the manuscript: How to implement a Clinical Ethics Committee in an Oncological Research Hospital? Qualitative results from a Process Evaluation Study using Normalization Process Theory (EVACEC).

This study highlight the Clinical Ethics Committee (CEC) of an Italian Local Health Authority (LHA) of Reggio Emilia, Italy, is a multi professional service established in 2020 to support healthcare professionals (HPs) in dealing with ethical issues in clinical practice. We evaluated the integration of the CEC into routine practice, 24 months after its implementation.

This manuscript is appropriate for publication by the PLOSONE JOURNAL It has the correct specifications, and approves :

1) Topics.

2) The quality of scientific paper and achieve publication goals, improve consistency and readability. Preserve flow of expression, Professional results, Improve structure and presentation, Plagiarism is check, also the Bibliographic editing, and Formatting finally improve the target of the journal.

3) All content presented in the manuscripts is perfect, and have keep the guidelines of journal, including opinions, Abstract, methods, statistical analysis and materials, results, images-figure especially the factor path diagram , discussions, and conclusions. The article is research work and provide new development of an Oncological Research Hospital, and Qualitative results from a Process Evaluation Study using Normalization Process Theory (EVACEC).

4) Manuscripts using a acceptable file formats use the layout options, on the length of the research articles, keep the main text of the body below appropriate words. Authors provide three separate documents during submission abstract, blinded text, figure 1 , tables with results, and highlights. Grammar and punctuation standards followed the content. References, keywords , figure provide in the text appendix also to the manuscript. International system of units be followed in the representation of all units in the text, recommended structure manuscript follows. References method formatting and style guidelines. Contribution to the literature: the uniqueness and importance of this work in terms of research topics, its methodologies, and outcomes, this article enhances the scholarship in the subject. Ideas are presented clear, concise, and complete manner. Manuscript is free from any repetitions, irrelevant information, or unjustified generalizations. Theoretical framework is explicitly stated. All claims are backed up with evidence and references. Research problems’ position and significance in existing literature is emphasized. Chosen methodology is suitable for the problem. Study’s findings are well presented with sufficient discussions and comparisons to existing literature. Objective and convincing measures exist to support the validity and reliability of the methodology and results. Relevant literature is properly cited, identified key components to support the normalization of CECs and enable their activation within a clinical setting. An active and sustainable CEC be visible, accessible, understood and trusted, clear in purpose, sufficiently integrated into the life of the organisation, adequately resourced, appropriately constituted and competent, accountable and independent. Findings can inform the development of practical strategies for CECs implementation and of appropriate outcomes for further evaluation.

And finally have complete the Aims and scope of the PLOS-ONE Journal.

I suggest: Please Accept the article as it is.

WITH REGARDS

the Reviewer

Reviewer #4: The investigators present a rigorous, scientifically sound method to qualitatively evaluate the impact of the implementation of a Clinical Ethics Committee into the operational setting of an oncological center in Italy. The proposed publication is the first in a series of publications that try to asesss the experiences of CEC members and managers who contributed to the set-up of this organism. his kind of research could be important in order to assess the relevance and impact of CEC implementation in the clinical setting. The study identifies some relevant concepts that participants perceive as important and pivotal to these kind of Committeess. The proposed research is methodologically sound and novel.

Notwithstanding, the following considerations should be taken into account:

1. The discussion should include a part of potential generalisation of study findings. Does the results and reflections extracted from this use case can be taken into account to future centres (oncological or not) that are considering implementing a Clinical Ethics Committee? Is this extrapolable to other countries, or only affects Italy? Please, provide in the discussion some further generalisibility reflections of the results.

2. The word "several" in line 506 has a typo.

Reviewer #5: The manuscript needs to be shorter; there are typos in the paper; too many abbreviations have been used, which is breaking the reader's flow; the target audience is too narrow due to the language used in the paper. The discussion section is longer than the results section which should not be the case as it is a research article. The methods section needs revision as well for more clarity.

6. PLOS authors have the option to publish the peer review history of their article (what does this mean? ). If published, this will include your full peer review and any attached files.

**Do you want your identity to be public for this peer review?** For information about this choice, including consent withdrawal, please see our Privacy Policy .

Reviewer #1: No

Reviewer #2: **Yes: ** Qosay Al-Balas

Reviewer #3: No

Reviewer #4: **Yes: ** Pau Alcubilla Prats

Reviewer #5: **Yes: ** Anamika

---

## [Author Response · Author response to Decision Letter 1]

20 Dec 2024

Dear editors, our manuscript has been revised according to your requests and it does follow journal submission guidelines now.

REVIEWER N#1

Thank you for your kind evaluation of our work and your precise comments. We really appreciated them, and we believe the revision you requested significantly improved our manuscript.

Thank you for your comment on whether the CEC was only for oncology; it provided us with the opportunity to elucidate a critical aspect of this service. We agreed that the paragraphs involving oncology were confusing. From their writing, it could be deduced that the CEC was dedicated only to oncologists and not–as it was instead–a service available for all healthcare professionals, no matter their clinical area.

Your comment was also an occasion to rethink the link with oncology throughout the manuscript. We have made multiple significant changes to the text to clarify this point and reformulated these concepts within the text. Also, we added a mention to the fact that we didn’t receive any request from oncologists in the limitation. Please, see: Background, lines 95-97; Strengths and limitations; lines 486-493).

Regarding your revision/suggestions, we made the following changes:

Title, grammar, punctuation:

• we removed the question mark;

• we revised the text, corrected all misspelling and punctuation errors, and fixed the formatting of the file as suggested;

• we welcomed all your suggestions on grammar and made the changes where applicable.

Abstract:

• we replaced the term “stakeholders” with “LHA directors” (line 24).

Background:

• We followed your instructions and revised the background accordingly: in fact, we first addressed CESS in general and then clarified that our study concerns a specific form of CESS (our CEC) (lines 39-61).

Methods:

• A pilot study was conducted with two participants. While no significant changes were necessary, adjustments and minor modifications were made based on the pilot responses (lines 137-138)

• we agreed it would benefit the reader to provide a brief description of the 4 NPT concepts within the main document and added it (lines 112-118)

• We revised the Methods section to include all the details of domains 1 and 2 of the COREQ to fill in all the missing information (lines 136-144). We did not write a separate “reflexivity” paragraph to avoid making the text less fluid. However, we believe the role of all team researchers and their relationship with the CEC and its members are now clear. We discuss potential biases in the Strengths and limitations section (lines 479-485).

• Yes, we developed two different topic-interviews guidelines: one for LDs and one for CMs. NPT concepts informed both and both are accessible in the Supplementary Materials.

• Thank you for suggesting combining some key areas together. They were indeed repetitive, so we revised them accordingly and synthesized them (lines 120-125).

• We have now explained in the manuscript how we found eligible participants and how we obtained consent (lines 122-125 and line 129)

• Since all the people involved in the study are native Italian speakers, all the interviews were conducted in Italian and were not translated. Also, all interviews were analyzed directly in Italian to respect the original meanings intended by the participants underlying their statements and to give voice to the semantic complexity of what emerged from the interviews. Only the quotations included in the manuscript were translated (translated by us) (lines 128-130)

• In the Data Analysis paragraph we referred to an excel file reporting all the data underlying the findings described in the manuscript, along with the codes from the qualitative analysis we conducted. That file was indeed missing in the first submission, we apologize. We have now uploaded it as Supplementary material (line 140).

• We added a phrase explaining how we managed conflicts and reached consensus, which was indeed missing (lines 145-147). Also, as specified in the text, the research team included a methodologist who helped structure the methodological section and reviewed the manuscript at each step.

Results:

• Thank you for commenting on Figure 1. Your comment helped us to better clarify an important aspect of our results. We agree with you, and we have amended the Figure accordingly.

• Since the president of the CEC is among the authors of the paper, we deemed it necessary to identify the president for transparency purposes regarding potential conflicts of interest or bias.

• We agree with the reviewer, and we confirm that the sentence in line 274 (now lines 282-285) is still part of our results. We modified the text to clear the sentence.

Discussion:

• We appreciate your compliment on how we addressed the humanization of care, which is a very crucial topic for us as well. Thank you very much. The humanization of care is a recurring theme in the bioethics literature; still, its explicit association with ethics is apparently only found in articles by Italian researchers (ref. 41 & 42 of the manuscript). We found no other records connecting the two concepts, and we hypothesize that this connection has something to do with the Italian cultural context. But it is, indeed, a hypothesis and we modified the text accordingly (line 397).

• We agree that our Discussion was too long, and that some concepts did not relate to the results section. We removed several phrases and paragraphs (see the track changes version).

• We agree that our use of CEC and CESS could have been clearer and revised the text to clarify it. We had to keep a reference to both CESS and CEC, as we refer multiple times to two main cross-sectional research lines (CESSs outcomes and CESSs implementation) but conducted a study on a CEC, a specific form of CESS. So, depending on the concept we were explaining, we had to refer generally to CESS or specifically to the object of our study. However, we believe the two concepts are more straightforward now and we better described which one we discussed in each paragraph (lines 378-382).

• We added comparisons to our results, as suggested (lines 409-412 and lines 422-424).

Practical implications:

• We greatly appreciate your comment on this section. In Table 4 caption, we have now clarified that its content was developed through a combination of our results and personal suggestions (see Table 4 caption).

REVIEWER N#2

Thank you very much for appreciating our work. Your questions allowed us to reflect on crucial aspects of our research and revise the manuscript accordingly.

Regarding how ethical issues were managed before the CEC: we clarified that the Bioethics Unit operated as the only ethics support service until the CEC was implemented and dealt with every request for ethics support/training (lines 75-76). Also, the CEC has been developed because of the experience gained by the Bioethics Unit ethics support services (lines 79-81)

As for the diversity of the CEC members, we did include one citizen’s representative as member, who was the Head of the Territorial Planning and Organization Area of CSV Emilia (Volunteer Service Center - CSV Emilia brings together volunteer and patient associations active in the provincial territory). However, it was explicitly stated in the CEC regulations that specific external experts—who can meaningfully represent the patient’s cultural and/or religious perspectives—could be invited by the CEC president during consultations, if necessary. We did not include those potential members in the sample, as they participate in the CEC discussion only occasionally, and do not offer a representative perspective on implementation (Methods, lines 122-125, Strengths and limitations, lines 494-496).

We concur that, due to the short time period under study, our findings can address the limitations of a CEC rather than establish a long-term impact; however, we believe our results are still valuable and useful for professionals who aim at performing similar evaluations.

Due to the CEC transition into a regional service, we cannot conduct a long-term or even longitudinal analysis of it, which would have shed more light on the implementation issue.

However, since the regional CEC (named COREC) was developed and implemented based on previous experience, it would be valuable to conduct an effectiveness evaluation of the COREC in the future.

While implemented within an oncological research hospital, the CEC was intended for healthcare professionals from all specialties within the institution, not only those working in cancer care (lines 95-97). Based on our results, we do not believe that every specialty should have its own specific CEC; rather, we believe that an effective CEC is well integrated into its context, and therefore, we think that each institution/hospital should have its own CEC for this to happen.

However, the CEC we studied has now become a regional committee (called COREC) for political reasons. Further research is needed to understand how a regional committee works and what is the best model.

We agree that patients' opinions (along with those of their loved ones) are pivotal in determining any changes the CEC would provide in healthcare services received. However, in this paper, we focused our evaluation on the implementation process. Our aim was to identify the components that make the CEC normalized and integrated in clinical practice for its users (healthcare professionals, HPs). We will include the perspectives of CEC users (HPs who requested an ethics consultation) in a dedicated paper.

We added some information about the ethics competence/training of the CEC members (lines 155-159).

REVIEWER #3

Thank you very much for your valuable contribution. The numerous and significant comments you have given on our work and your positive feedback have been a great source of encouragement for our team and motivate to continue our research with dedication.

REVIEWER #4

Thank you for your interest in our work and your kind comments. We appreciate your evaluation of our manuscript. We welcomed your suggestions and, accordingly, made a few changes to the Discussion section. We fixed all typos and misspelling errors.

We had already addressed the potential generalization of our Results; in fact, Table 4 (§ “Practical implications”) was built with that aim in mind. However, we had not explicitly mentioned potential generalization nor was the distinction between results and original proposals clear, so we added a short paragraph on the possible generalization of our results (lines 466-468).

REVIEWER #5

Thank you for your comments. We agree our manuscript was too long. We significantly shortened the manuscript whenever possible and removed all the content not directly related to the described results from the Discussion. We fixed all typos, tried to keep abbreviations to a minimum and spelled them out in full whenever possible (e.g. Bioethics Unit instead of BU). We revised the Methods section for clarity, as suggested. Thank you for pointing out that our language was restrictive. We made efforts to simplify it and believe our manuscript is now more readily understandable.

---

## [Decision Letter · Decision Letter 1]

23 Jan 2025

How to implement a Clinical Ethics Committee in an Oncological Research Hospital: qualitative results from a Process Evaluation Study using Normalization Process Theory (EVACEC).

PONE-D-24-40359R1

Dear Dr. Crico,

We’re pleased to inform you that your manuscript has been judged scientifically suitable for publication and will be formally accepted for publication once it meets all outstanding technical requirements.

Kind regards,

Alejandro Botero Carvajal, MD

Academic Editor

PLOS ONE

Additional Editor Comments (optional):

Reviewers' comments:

Reviewer's Responses to Questions

**Comments to the Author**

1. If the authors have adequately addressed your comments raised in a previous round of review and you feel that this manuscript is now acceptable for publication, you may indicate that here to bypass the “Comments to the Author” section, enter your conflict of interest statement in the “Confidential to Editor” section, and submit your "Accept" recommendation.

Reviewer #1: All comments have been addressed

2. Is the manuscript technically sound, and do the data support the conclusions?

Reviewer #1: Yes

3. Has the statistical analysis been performed appropriately and rigorously? 

Reviewer #1: Yes

4. Have the authors made all data underlying the findings in their manuscript fully available?

Reviewer #1: Yes

5. Is the manuscript presented in an intelligible fashion and written in standard English?

Reviewer #1: Yes

6. Review Comments to the Author

Reviewer #1: The authors have reviewed all my comments and I thank them for taking my suggestions into consideration.

The manuscript has been significantly improved.

I would suggest they give it a last read to correct some very minor punctuation errors (Line 119 missing point, line 130 extra space ...).

7. PLOS authors have the option to publish the peer review history of their article (what does this mean? ). If published, this will include your full peer review and any attached files.

**Do you want your identity to be public for this peer review?** For information about this choice, including consent withdrawal, please see our Privacy Policy .

Reviewer #1: No

---

## [Editor Report · Acceptance letter]

PONE-D-24-40359R1

PLOS ONE

Dear Dr. Crico,

I'm pleased to inform you that your manuscript has been deemed suitable for publication in PLOS ONE. Congratulations! Your manuscript is now being handed over to our production team.

Kind regards,

on behalf of

Dr. Alejandro Botero Carvajal

Academic Editor

PLOS ONE